# SHADE: SHANNON DECAY INFORMATION-BASED REGULARIZATION FOR DEEP LEARNING

## ABSTRACT

Regularization is a big issue for training deep neural networks. In this paper, we propose a new information-theory-based regularization scheme named SHADE for SHAnnon DEcay. The originality of the approach is to define a prior based on conditional entropy, which explicitly decouples the learning of invariant representations in the regularizer and the learning of correlations between inputs and labels in the data fitting term. We explain why this quantity makes our model able to achieve invariance with respect to input variations. We empirically validate the efficiency of our approach to improve classification performances compared to standard regularization schemes on several standard architectures.

## 1 INTRODUCTION

Deep neural networks (DNNs) have shown impressive state-of-the-art results in the last years on numerous tasks and especially for image classification (Krizhevsky et al., 2012; He et al., 2016). One key element is the use of very deep models with a huge number of parameters. Accordingly, DNNs need to be trained on a lot of data (*e.g.* ImageNet). Regularization methods such as weight decay (Krogh & Hertz, 1992), dropout (Srivastava et al., 2014) or batch normalization (Ioffe & Szegedy, 2016) are common practice to mitigate the ratio between the numbers of training samples and model parameters. Despite constant progress of DNNs performance, their generalization ability is still largely misunderstood and theoretical tools such as PAC-based analysis seem limited to explain it. Meanwhile, the question of DNN regularization remains open as demonstrated by Zhang et al. (2017).

Information Bottleneck (IB) is an interesting framework to address regularization (Tishby et al., 1999). It suggests that it is possible to compare the generalization ability of different models by studying the compression rate of their representations: a model that is able to remove more information from the input while providing enough information about the label is more likely to generalize well. Considering an input variable $X$, label $C$ and its deep representation $Y = h(X)$, IB regularizes the training by minimizing the mutual information $I(X, Y)$ at constant mutual information $I(C, Y)$. Along with IB also come some theoretical investigations, with the definition of generalization bounds in Shamir et al. (2010).

A challenge for applying IB principles to DNNs is the definition of information-based regularizers that are differentiable and easily computable in order to be efficiently trained using back-propagation (LeCun et al., 1998). There are recent attempts in this direction. For example, Alemi et al. (2017) proposes a DNN that predicts the parameters of the distribution of the random representation variables conditionally to the input and use it to compute IB measures with variational bounds. Achille & Soatto (2016) model the information about the data in the learned weights and propose to minimize it.

In this paper, we propose a new regularization loss minimizing $H(Y \mid C)$, the entropy of the representation variable *conditionally* to the label variable, instead of the mutual information $I(X, Y)$. For a deterministic model, meaning that the entropy $H(Y \mid X)$ is null, we have $I(X, Y) = I(C, Y) + H(Y \mid C)$[1]. By minimizing directly $H(Y \mid C)$ we explicitly decouple the data fitting term $I(C, Y)$, which attempts at finding correlations between inputs and labels, and a term which represents the compression rate of the representation $H(Y \mid C)$.

---

[1]For any two variables $U, V$ we have $I(U, V) = H(U) - H(U \mid V) = H(V) - H(V \mid U)$.

Our contributions are three-fold. First, we derive in Section 2 a new regularization scheme, called SHADE for SHAnnon DEcay, to tackle the difficulty of getting an accurate, differentiable and scalable estimate of $H(Y \mid C)$. It is based on a latent code $Z$ representing the activation state of a neuron $Y$, allowing us to express a complete tractable loss with few additional computation. Second, we discuss the interest of our regularization framework based on conditional entropy in Section 3. In particular, we argue that this measure $H(Y \mid C)$ is well designed to compare representations in terms of invariance while still taking the target task into account. Finally, we extensively experiment our SHADE regularizer on different deep learning models and several challenging datasets (CIFAR-10, ImageNet, MNIST-M) in Section 4.

## 2 SHADE: A NEW REGULARIZATION METHOD BASED ON $H(Y \mid C)$

**Notations and definitions.** $X$ is the input from input space $\mathcal{X}$ with high dimensionality and $C$ is the class label from finite class space $\mathcal{C} = \{1, 2, .., |\mathcal{C}|\}$. A hypothesis space is designed with a family of parametric (continuous) functions $\mathcal{H} = \{h(w, .) : \mathcal{X} \to \mathcal{Y}; \ w \in \mathcal{W} = \mathbb{R}^m\}$. A model $h \in \mathcal{H}$ outputs a representation of the input from which a class is predicted. In many tasks like image classification, the representation $Y = h(X)$ of an input $X$ usually defines a distribution, noted $q_{Y|X}$, on the class space (*e.g.* obtained by applying a softmax function). The training in supervised learning consists in finding the parameters $w$ of $h$ minimizing the expectancy of a loss function $\mathcal{L}_{\mathrm{cls}}(Y, C)$ that quantifies the difference between the prediction and the label. It is usually the cross-entropy between the obtained distribution $q_{Y|X}$ and the one-hot distribution with unit probability for the class $C$ noted $p_C$: $\ell_{\mathrm{cls}}(Y, C) = -\sum_{c \in \mathcal{C}} p_C(c) \log\big(q_{Y|X}(c)\big)$. A regularization term $\Omega(\cdot)$ with a coefficient $\beta$ is often added to the classification loss, aiming at influencing the learning toward models with lower complexity[2]. In image classification, the most commonly used regularization criterion is the weight decay: $\Omega(w) = ||w||_2^2$. In the IB framework, the regularization term would be $\Omega(w) = I\big(h(w, X), X\big)$. The final loss for supervised learning is:

$$\mathcal{L} = \mathbb{E}_{(Y,C)}\big(\ell_{\mathrm{cls}}(Y, C) + \beta\Omega(w)\big). \tag{1}$$

In this section, we will further describe SHADE, a new regularization term based on the conditional entropy $H(Y \mid C)$ designed to drive the optimization toward more invariant representation.

### 2.1 ENTROPY-BASED REGULARIZATION FOR DEEP NEURAL NETWORKS

**Layer-wise regularization.** A DNN is composed of a number $L$ of layers that transform sequentially the input. Each one can be seen as an intermediate representation variable, noted $Y_\ell$ for layer $\ell$, that is determined by the output of the previous layer and a set of parameters $\mathbf{w}_\ell$. Each layer filters out a certain part of the information from the initial input. Thus, from the data processing inequality in Cover & Thomas (1991) can be derived the following inequalities for any $\ell$:

$$H(Y_\ell \mid C) \le H(Y_{\ell-1} \mid C) \le ... \le H(Y_1 \mid C) \le H(X \mid C). \tag{2}$$

Similarly to the recommendation of Tishby & Zaslavsky (2015), we apply regularization to all the layers, using a layer-wise criterion $H(Y_\ell \mid C)$, and producing a global criterion:

$$\Omega_{\mathrm{layers}} = \sum_{\ell=1}^{L} H(Y_\ell \mid C). \tag{3}$$

**Unit-wise regularization.** Examining one layer $\ell$, its representation variable is a random vector of coordinates $Y_{\ell,i}$ and of dimension $D_\ell$: $Y_\ell = (Y_{\ell,1}, ..., Y_{\ell,D_\ell})$. The upper bound[3] $H(Y_\ell \mid C) \le \sum_{i=1}^{D_\ell} H(Y_{\ell,i} \mid C)$ enables to define a unit-wise criterion that SHADE seeks to minimize. For each

---

[2]There are other methods of regularization such as dropout (Srivastava et al., 2014) or data augmentation (van Dyk & Meng, 2001) that do not apply directly to the loss.

[3]This upper bound is well justified in deep learning as the neurons of a layer tend to be more and more independent of each other as we go deeper within the network.

unit $i$ of every layer $\ell$ we design a loss $\omega_{\mathrm{unit}}(Y_{\ell,i} \mid C) = H(Y_{\ell,i} \mid C)$ that will be part of the global regularization loss:

$$\Omega_{\mathrm{layers}} \leq \Omega_{\mathrm{units}} = \sum_{l=1}^{L} \sum_{i=1}^{D_\ell} \underbrace{H(Y_{\ell,i} \mid C)}_{\omega_{\mathrm{unit}}(Y_{\ell,i}|C)}. \tag{4}$$

Later in the paper, we use the notation $Y$ instead of $Y_{\ell,i}$ for simplicity since the coordinates are all considered independently to define $\omega_{\mathrm{unit}}(Y_{\ell,i} \mid C)$.

## 2.2 ESTIMATING ENTROPY

In this section, we describe how to define a loss based on the measure $H(Y \mid C)$ with $Y$ being one coordinate variable of one layer. Defining this loss is not obvious as the gradient of $H(Y \mid C)$ with respect to the layer's parameters may be computationally intractable. $Y$ has an unknown distribution and without modeling it properly it is not possible to compute $H(Y \mid C)$ precisely for the following reasons.

Since $H(Y \mid C) = \sum_{c \in \mathcal{C}} p(c) H(Y \mid c)$ it is necessary to compute $|\mathcal{C}|$ different entropies $H(Y \mid c)$. This means that, given a batch, the number of samples used to estimate one of these entropies is divided by $|\mathcal{C}|$ on average which becomes particularly problematic when dealing with a large number of classes such as the 1,000 classes of ImageNet. Furthermore, entropy estimators are extremely inaccurate considering the number of samples in a batch. For example, LME estimators of entropy in Paninski (2003) converge in $\mathcal{O}(^{(\log K)^2}\!/_K)$ for $K$ samples. Finally, most estimators such as LME require discretizing the space in order to approximate the distribution *via* a histogram. This raises issues on the bins definition considering that the variable distribution is unknown and varies during the training in addition to the fact that having a histogram for each neuron is computationally and memory consuming.

To tackle these drawbacks we investigate the two following tricks: the introduction of a latent variable that enables to use more examples to estimate the entropy; and a bound on the entropy of the variable by an increasing function of its variance to avoid the issue of entropy estimation with a histogram and make the computation tractable and scalable.

**Latent code.**  First, inspecting a neuron $Y$ (before ReLU), the ReLU activation makes it act as a detector, returning a signal when a certain pattern is present on the input. If the pattern is absent the signal is zero, otherwise, it quantifies the resemblance with it. We therefore propose to associate a binomial variable $Z$ with each unit variable $Y$ (before ReLU). This variable $Z$ indicates if a particular pattern is present on the input ($Z = 1$ when $Y \gg 0$) or not ($Z = 0$ when $Y \ll 0$). It acts like a latent code in variational models (*e.g.* Kingma & Welling, 2014) or in generative models (*e.g.* Chen et al., 2016). In our implementation, we chose a binomial distribution $p(Z = 1 \mid Y) = \mathrm{sigmoid}(Y)$ that matches this intuition, as detailed in Sec. 2.3.

Furthermore, it is very likely that most intermediate features of a DNN can take similar values for inputs of different classes – this is especially true for low-level features. The semantic information provided by a feature is thus more about a particular pattern than about the class itself. Only the association of features allows determining the class. So $Z$ represents a semantically meaningful factor about the class $C$ and from which the input $X$ is generated. The feature value $Y$ is then a quantification of the possibility for this semantic attribute $Z$ to be present in the input or not.

Then, we assume the Markov chain $C \to Z \to X \to Y$. Indeed, during the training, the distribution of $Y$ varies in order to get as close as possible to a sufficient statistic of $X$ for $C$ (see definition in Cover & Thomas, 1991). Therefore, we expect $Z$ to be such that $Y$ draws near a sufficient statistic of $Z$ for $C$ as well. By assuming the sufficient statistic relation $I(Y, C) = I(Y, Z)$ we get the equivalent equality $H(Y \mid C) = H(Y \mid Z)$, and finally obtain:

$$\omega_{\mathrm{unit}}(Y \mid C) = H(Y \mid C) = H(Y \mid Z) = \sum_{z \in \{0,1\}} p(z) H(Y \mid Z = z). \tag{5}$$

This modeling of $Z$ as a binomial variable (one for each unit) has the advantage of enabling good estimators of conditional entropy since we only divide the batch into two sets for the estimation ($z = 0$ and 1) regardless of the number of classes.

---

**Algorithm 1** Moving average updates: for $z \in \{0, 1\}$, $p^z$ estimates $p(Z = z)$ and $\mu^z$ estimates $\mathbb{E}(Y \mid Z = z)$

---

1: **Initialize:** $\mu^0 = -1$, $\mu^1 = 1$, $p^0 = p^1 = 0.5$, $\lambda = 0.8$
2: **for each** mini-batch $\{y^{(k)}, k \in 1..K\}$ **do**
3:      **for** $z \in \{0, 1\}$ **do**
4:          $p^z \leftarrow \lambda p^z + (1 - \lambda) \frac{1}{K} \sum_{k=1}^{K} p(z \mid y^{(k)})$
5:          $\mu^z \leftarrow \lambda \mu^z + (1 - \lambda) \frac{1}{K} \sum_{k=1}^{K} \frac{p(z \mid y^{(k)})}{p^z} y^{(k)}$
6:      **end for**
7: **end for**

---

**Variance bound.** The previous trick allows computing fewer entropy estimates to obtain the global conditional entropy, thus increasing the sample size used for each entropy estimation. Unfortunately, it does not solve the bin definition issue. To address this, we propose to use the following bound on $H(Y \mid Z)$, that does not require the definition of bins:

$$H(Y \mid Z) \leq \frac{1}{2} \ln \left( 2\pi e \mathbb{V}\mathrm{ar}(Y \mid Z) \right). \tag{6}$$

This bound holds for any continuous distributions $Y$ and there is equality if the distribution is Gaussian. For many other distributions such as the exponential one, the entropy is also equal to an increasing function of the variance. In addition, one main advantage is that variance estimators are much more robust than entropy estimators, converging in $\mathcal{O}(1/K)$ for $K$ samples instead of $\mathcal{O}(\log(K)^2/K)$.

Finally, the $\ln$ function being one-to-one and increasing, we only keep the simpler term $\mathbb{V}\mathrm{ar}(Y \mid Z)$ to design our final loss:

$$\Omega_{\mathrm{SHADE}} = \sum_{\ell=1}^{L} \sum_{i=1}^{D_\ell} \sum_{z \in \{0,1\}} p(Z_{\ell,i} = z \mid Y) \mathbb{V}\mathrm{ar}(Y \mid Z_{\ell,i} = z). \tag{7}$$

In next section, we detail the definition of the differential loss using $\mathbb{V}\mathrm{ar}(Y \mid Z)$ as a criterion computed on a mini-batch.

## 2.3 INSTANTIATING SHADE

For one unit of one layer, the previous criterion writes:

$$\mathbb{V}\mathrm{ar}(Y \mid Z) = \int_{\mathcal{Y}} p(y) \int_{\mathcal{Z}} p(z \mid y) \big(y - \mathbb{E}(Y \mid z)\big)^2 \, \mathrm{d}z \, \mathrm{d}y \tag{8}$$

$$\approx \frac{1}{K} \sum_{k=1}^{K} \left[ \int_{\mathcal{Z}} p(z \mid y^{(k)}) \big(y^{(k)} - \mathbb{E}(Y \mid z)\big)^2 \, \mathrm{d}z \right]. \tag{9}$$

The quantity $\mathbb{V}\mathrm{ar}(Y \mid Z)$ can be estimated with Monte-Carlo sampling on a mini-batch of input-target pairs $\big\{(x^{(k)}, c^{(k)})\big\}_{1 \leq k \leq K}$ of intermediate representations $\big\{y^{(k)}\big\}_{1 \leq k \leq K}$ as in Eq. (9).

$p(Z \mid y)$ interpreted as the probability of presence of attribute $Z$ on the input, it should clearly be modeled such that $p(Z = 1 \mid y)$ increases with $y$. The more similarities between the input and the pattern represented by $y$, the higher the probability of presence for $Z$. We suggest using:

$$p(Z = 1 \mid y) = \sigma(y) \qquad p(Z = 0 \mid y) = 1 - \sigma(y) \qquad \text{with sigmoid function } \sigma(y) = \frac{1}{1 + e^{-y}}.$$

For the expected values $\mu^z = \mathbb{E}(Y \mid z)$ we use a classic moving average that is updated after each batch as described in Algorithm 1. Note that the expected values are not changed by the optimization since they have no influence on the entropy $H(Y \mid Z)$.

The concrete behavior of SHADE can be interpreted by analyzing its gradient as described in Appendix C.

For this proposed instantiation, our SHADE regularization penalty takes the form:

$$\Omega_{\text{SHADE}} = \sum_{\ell=1}^{L} \sum_{i=1}^{D_\ell} \sum_{k=1}^{K} \sum_{z \in \{0,1\}} p\left(Z_{\ell,i} = z \,\Big|\, y_{\ell,i}^{(k)}\right) \left(y_{\ell,i}^{(k)} - \mu_{\ell,i}^{z}\right)^2. \tag{10}$$

We have presented a regularizer that is applied layer-wise and that can be integrated into the usual optimization process of a DNN. The additional computation and memory usage induced by SHADE is almost negligible (computation and storage of two moving averages per neuron). Namely, SHADE adds half as many parameters as batch normalization does.

## 3 CONDITIONAL ENTROPY AS INVARIANCE MEASURE FOR CLASSIFICATION

Inspired by the IB framework, we have derived a criterion that SHADE loss aims at minimizing. In this section, we further explore the relevance of our conditional entropy term for regularization. We first establish a connection between the entropy $H(Y)$ of a representation and its invariance. Then we explain why $H(Y \mid C)$ seems to be a better fitting criterion for a classification task.

**$H(Y)$ to measure invariance.** For an input $X$ and a deterministic mapping $f(X) = Y$, we have $H(Y \mid X) = 0$ and therefore:

$$H(Y) = I(X, Y) + H(Y \mid X) = I(X, Y) = H(X) - H(X \mid Y). \tag{11}$$

$H(X)$ being fixed, $H(Y)$ is inversely related to $H(X \mid Y)$. Besides, we assume that $H(X \mid Y)$ is a good measure to quantify how invariant a representation is. Indeed, if a representation is invariant to many transformations, many inputs have the same representation. Consequently, given a representation sample, it is difficult to guess from which input it has been computed. These properties are perfectly captured by $H(X \mid Y)$, representing the uncertainty in $X$ knowing $Y$. The bigger the uncertainty, the harder it is to predict $X$ precisely. Concretely, when trying to guess $X$ knowing $Y$, we can lower bound the error made in the best case, with an increasing function of the conditional entropy as developed in Appendix A. Therefore, it seems that $H(Y)$ is a good measure of the invariance of the model.

In the particular case of deep learning, He et al. (2016) explain that the stacking of multiple layers is responsible for improving the generalization of DNN. This fact can be explained by the data processing inequality (Cover & Thomas, 1991). In the case of finite input space, each layer is responsible for filtering a certain amount of information. As clearly illustrated in Tishby & Zaslavsky (2015), for each stage, the representation has a lower entropy than the representation of the preceding layer. Increasing the depth increases the capacity of the network to reduce the overall entropy of the DNN representation thus increasing their invariance.

**$H(Y)$ as a regularizer.** From this intuition, one can use a variance bound similar to Eq. (6): $H(Y) \leq 1/2 \ln(2\pi e \mathbb{V}\text{ar}(Y))$ to derive a loss in order to minimize the representations' entropy $H(Y)$. For instance, we can show that the weight decay reduces this variance under some approximations. In fact when $Y = \mathbf{w}^\top X + \mathbf{b}$, by estimating $\Lambda = \mathbb{C}\text{ov}(X)$, the variance takes the immediate form $\mathbb{V}\text{ar}(Y) = \mathbf{w}^\top \Lambda \mathbf{w}$. If $\Lambda = Id$, meaning that the input coordinates are considered independent with unit variance, then $\mathbb{V}\text{ar}(Y) = ||\mathbf{w}||_2^2$. It corresponds to the weight decay regularization or $L_2$ penalty. Even if within a DNN layer, the batch normalization tends to enforce this unit variance hypothesis and the depth of DNN tends to ensure the independence hypothesis the weight decay remains poor at improving generalization as illustrated in Zhang et al. (2017). Considering fewer assumptions, it is possible to reduce the variance of an output coordinate by penalizing its empirical variance on a batch during the training. We introduce an alternative loss on the variance called VarEntropy, constructed the same way SHADE has been derived, but avoiding the introduction of a latent variable $Z$:

$$\Omega_{\text{VarEntropy}} = \frac{1}{K} \sum_{k=1}^{K} \left(y^{(k)} - \mathbb{E}(Y)\right)^2. \tag{12}$$

Even if experimentally, VarEntropy regularization seems to behave better than simple weight decay, it is not as efficient as SHADE and we try to explain why below.

**Importance of conditional entropy.** We have seen that lowering the entropy of the representation enables to make it more invariant. However, another fact reported by He et al. (2016) is that stacking layers increases the difficulty to train the network. Indeed, when reducing too much the entropy, there is a risk that the information about the label is filtered as well. The representation is so invariant that it is no longer possible to distinguish between the classes. In He et al. (2016), they solve this issue by forcing the transmission of additional information through skip-connections while IB prescribes to maximize the compression rate at constant information about the label. All this highlights the fact that having invariant representations is interesting if it is intra-class invariant. Since we do not want two inputs from different classes to have the same representation, we prefer to focus on a criterion quantifying the intra-class compression rate in order to maximize intra-class invariance: $H(Y \mid C)$.

As mentioned in the introduction, $H(Y \mid C)$ thus differs from standard IB frameworks based on $I(X, Y)$ (Achille & Soatto, 2016; Alemi et al., 2017). Contrarily to the mutual information $I(X, Y) = H(Y \mid C) + I(C, Y)$, our regularizer applied to $H(Y \mid C)$ ignores $I(C, Y)$. When minimizing $I(X, Y)$, there is no control on how both terms $H(Y \mid C)$ and $I(C, Y)$ are modified. Our regularizer is therefore beneficial in the sense that minimizing $H(Y \mid C)$ does not conflict with the mutual information $I(C, Y)$ between the representation and the label, information that is therefore for classification and should not be penalized.

## 4 EXPERIMENTS

### 4.1 IMAGE CLASSIFICATION WITH VARIOUS ARCHITECTURES ON CIFAR-10

Table 1: Classification accuracy (%) on CIFAR-10 test set.

|              | MLP     | AlexNet   | ResNet  | Inception |
|--------------|---------|-----------|---------|-----------|
| No regul.    | 62.38   | 83.25     | 89.84   | 90.97     |
| Weight decay | 62.69   | 83.54     | 91.71   | 91.87     |
| Dropout      | 65.37   | 85.95     | 89.94   | 91.11     |
| VarEntropy   | 63.70   | 83.61     | 91.72   | 91.83     |
| SHADE        | 66.05   | 85.45     | **92.15** | **93.28** |
| SHADE+D      | **66.12** | **86.71** | 92.03   | 92.51     |

We perform image classification on the CIFAR-10 dataset, which contains 50k training images and 10k test images of $32 \times 32$ RGB pixels, fairly distributed within 10 classes (see Krizhevsky, 2009, for details). Following the architectures used in Zhang et al. (2017), we use a small Inception model, a three-layer MLP, and an AlexNet-like model with 3 convolutional and 2 fully connected layers. We also use a ResNet architecture from Zagoruyko & Komodakis (2016). Those architectures represent a large family of DNN and some have been well studied in Zhang et al. (2017) within the generalization scope. For training, we use randomly cropped images of size $28 \times 28$ with random horizontal flips. For testing, we simply center-crop $28 \times 28$ images. We use momentum SGD for optimization (same protocol as Zhang et al., 2017).

We compare SHADE with three regularization methods, namely *weight decay*, *dropout* and *VarEntropy* presented in Eq. (12). For all architectures, the regularization parameters have been cross-validated to find the best ones for each method and the obtained accuracies on the test set are reported in Table 1.

We obtain the same trends as Zhang et al. (2017), which get a small improvement of 0.31% with weight decay on AlexNet. The improvement with weight decay is slightly more important with ResNet and Inception (0.87% and 0.90%) probably thanks to the use of batch normalization. In our experiments dropout improves generalization performances only for AlexNet and MLP. It is known that the use of batch normalization lowers the benefit of dropout, which is in fact not used in He et al. (2016).

We first notice that for all kind of architectures the use of SHADE significantly improves the generalization performance. It demonstrates the ability of SHADE to regularize the training of deep architectures. Moreover, SHADE systematically outperforms other regularizations of the same type

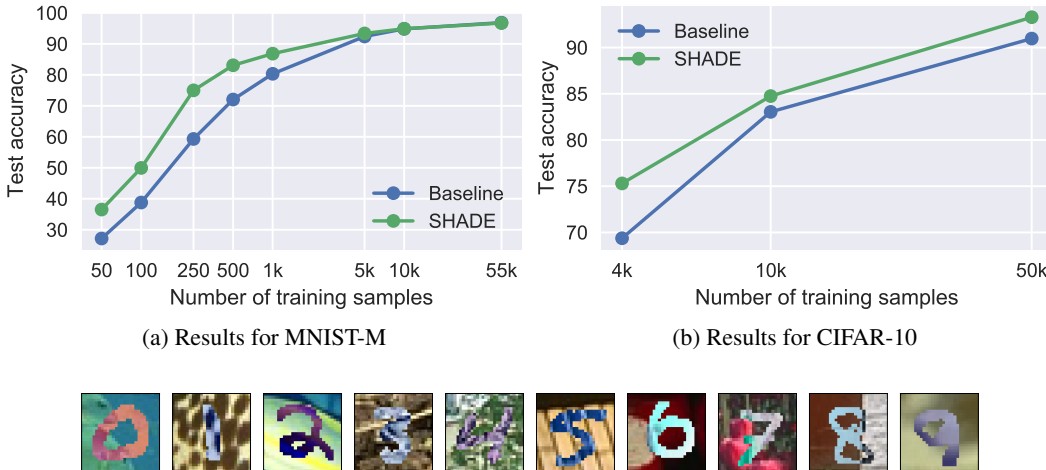

(a) Results for MNIST-M

(b) Results for CIFAR-10

(c) Examples of MNIST-M images misclassified by the baseline and correctly classified using SHADE, both trained with 250 samples.

Figure 1: Results when training with a limited number of samples in the training set for MNIST-M and CIFAR-10 with and without SHADE.

such as weight decay or VarEntropy, illustrating the advantage of minimizing the conditional entropy instead of the entropy directly.

Finally, SHADE shows better performances than dropout on all architecture except on AlexNet, for which they seem to be complementary, probably because of the very large number of parameters in the fully-connected layers, with best performances obtained with SHADE coupled with dropout (named SHADE+D). This association is also beneficial for MLP. On Inception and ResNet, even if dropout and SHADE independently improve generalization performances, their association is not as good as SHADE alone, probably because it enforces too much regularization.

## 4.2 LARGE SCALE CLASSIFICATION ON IMAGENET

In order to experiment SHADE regularization on very large scale dataset, we train on ImageNet (Russakovsky et al., 2015) a WELDON network from Durand et al. (2016) adapted from ResNet-101. This architecture changes the forward and pooling strategy by using the network in a fully-convolutional way and adding a max+min pooling, thus improving the performance of the baseline network. We used the **pre-trained weights of ResNet-101** (from the torchvision package of PyTorch) giving performances on the test set of **77.56%** for top-1 accuracy and 93.89% for top-5 accuracy. Provided with the pre-trained weights, the **WELDON architecture** obtains **78.51%** for top-1 accuracy and 94.65% for top-5 accuracy. After fine tuning the network using **SHADE** for regularization we finally obtained **80.14%** for top-1 accuracy and 95.35% for top-5 accuracy for a concrete improvement. This demonstrates the ability to apply SHADE on very large scale image classification successfully.

## 4.3 TRAINING WITH A LIMITED NUMBER OF SAMPLES

When datasets are small, DNNs tend to overfit quickly and regularization becomes essential. Because it tends to filter out information and make the network more invariant, SHADE seems to be well fitted for this task. To investigate this, we propose to train DNNs with and without SHADE on CIFAR-10 and MNIST-M with different numbers of samples in the training set.

First, we tested this approach on the digits dataset MNIST-M (Ganin & Lempitsky, 2015). This dataset consists of the MNIST digits where the background and digit have been replaced by colored and textured information (see Fig. 1c for examples). The interest of this dataset is that it contains lots of unnecessary information that should be filtered out, and is therefore well adapted to measure the effect of SHADE. A simple convolutional network has been trained with different numbers of samples of MNIST-M and the optimal regularization weight for SHADE have been determined on

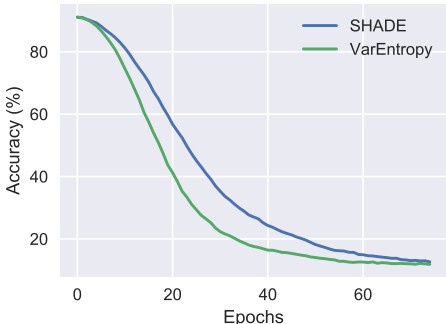

Figure 2: Evolution of the accuracy of a pre-trained Inception model on CIFAR-10 when only applying regularization.

the validation set (see training details in Appendix D). The results can be seen on Figure 1a. We can see that especially for small numbers of training samples ($< 1000$), SHADE provides an important gain of 10 to 15% points over the baseline. This shows that SHADE helped the model in finding invariant and discriminative patterns using less data samples.

Additionally, Figure 1c shows samples that are misclassified by the baseline model but correctly classified when using SHADE. These images contain a large amount of intra-class variance (color, texture, etc.) that is not useful for the classification tasks, explaining why adding SHADE, that encourages the model to discard information, allows important performance gains on this dataset and especially when only few training samples are given.

Finally, to confirm this behavior, we also applied the same procedure in a more conventional setting by training an Inception model on CIFAR-10. Figure 1b shows the results in that case. We can see that once again SHADE helps the model gain in performance and that this behavior is more noticeable when the number of samples is limited, allowing a gain of 6% when using 4000 samples.

### 4.4   CLASS-INFORMATION PRESERVATION BY CONDITIONAL ENTROPY

The main difference between SHADE and VarEntropy is the introduction of the latent variable $Z$, supposed to contain the information about the label $C$. The motivation of this extension is that minimizing $H(Y \mid C)$ instead of $H(Y)$ enables to save the class information during the optimization of the regularization loss, as explained in Section 3. To illustrate this benefit, we compare the impact of the two regularization losses on the classification performances of a pre-trained model. To do so, we fine tune a pre-trained Inception model only with a regularization loss, either based on $\mathbb{V}\mathrm{ar}(Y)$ or $\mathbb{V}\mathrm{ar}(Y \mid Z)$; without any label data or classification loss. The network performance obviously declines for both regularizers as we can see on Figure 2. However, this decline is slower with SHADE than VarEntropy. This confirms the intuition that $Z$ contains class-information and that SHADE produces less class-information filtering. This is explained by the optimization of the metric $\mathbb{V}\mathrm{ar}(Y \mid Z)$ that uses implicitly-learned information encoded in the network.

## 5   CONCLUSION

In this paper, we introduced a new regularization method for DNNs training, SHADE, which focuses on minimizing the entropy of the representation conditionally to the labels. This regularization aims at increasing the intra-class invariance of the model while keeping class information. SHADE is tractable, adding only a small computational overhead when included into an efficient SGD training. We show that our SHADE regularization method significantly outperforms standard approaches such as weight decay or dropout with various DNN architectures on CIFAR-10. We also validate the scalability of SHADE by applying it on ImageNet. The invariance potential brought out by SHADE is further illustrated by its ability to ignore irrelevant visual information (texture, color) on MNIST-M. Finally, we also highlight the increasing benefit of our regularizer when the number of training examples becomes small.

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

## A  ENTROPY BOUNDING THE RECONSTRUCTION ERROR

In section 3 we highlight a link between the entropy $H(X \mid Y)$ and the difficulty to recover the input $X$ from its representation $Y$. Here we exhibit a concrete relation between the reconstruction error, that quantifies the error made by a strategy that predicts $X$ from $Y$, and the conditional entropy. This relation takes the form of an inequality, bounding the error measure in the best case (with the reconstruction strategy that minimizes the error) by an increasing function of the entropy. We note $\hat{x}(Y) \in \mathcal{X}$ the reconstruction model that tries to guess $X$ from $Y$.

**The discrete case**  In case the input space is discrete, we consider the zero-one reconstruction error for one representation point $Y$: $\varepsilon(Y) = E_X[\mathbb{1}(\hat{x}(Y) \neq X)]$. This is the probability of error when predicting $\hat{x}(Y)$ from $Y$. The function that minimizes the expected error $\mathcal{E} = \mathbb{E}_Y(\varepsilon(Y))$ is $\hat{x}(Y) = \arg\max_{x \in \mathcal{X}} p(x \mid Y)$ as shown in Proof 1. We derive the following inequality:

$$\frac{H(X \mid Y) - 1}{\log |\mathcal{X}|} \leq \mathcal{E} \leq 1 - 2^{-H(X \mid Y)}. \tag{13}$$

The left side of the inequality uses Fano's inequality in Cover & Thomas (1991), the right one is developed in proof 2. This first bound shows how the reconstruction error and the entropy are related. For very invariant representations, it is hard to recover the input from $Y$ and the entropy of $H(X \mid Y)$ is high.

Besides, there can be an underlying continuity in the input space and it could be unfair to penalize predictions close to the input as much as predictions far from it. We expose another case below that takes this proximity into account.

**The continuous case**  In the case of convex input space and input variable with continuous density, we consider the 2-norm distance as reconstruction error: $\varepsilon(Y) = \mathbb{E}_{X \mid Y}\big[||X - \hat{X}(Y)||_2^2\big]$. This error penalizes the average distance of the input and its reconstruction from $Y$. The function that minimizes the expected error $\mathcal{E} = \mathbb{E}_Y[\varepsilon(Y)]$ is the conditional expected value: $\hat{x}(y) = \mathbb{E}[X \mid Y = y]$. Then $\mathcal{E} = \mathrm{Var}(X \mid Y)$. Helped by the well-known inequality $H(X \mid Y) \leq \frac{1}{2} \ln(2\pi e \mathrm{Var}(X \mid Y))$ we obtain:

$$\frac{e^{2H(X \mid Y)}}{2\pi e} \leq \mathcal{E}. \tag{14}$$

Here again, notice that a high entropy $H(X \mid Y)$ implies a high reconstruction error in the best case.

### A.1  PROOF 1

We have

$$\mathcal{E} = \int_{\mathcal{Y}} p(y)\varepsilon(y)\,\mathrm{d}y \tag{15}$$

$$= \int_{\mathcal{Y}} p(y)p(X \neq \hat{x}(y) \mid y)\,\mathrm{d}y \tag{16}$$

$$= \int_{\mathcal{Y}} p(y)(1 - p(\hat{x}(y) \mid y))\,\mathrm{d}y. \tag{17}$$

Since

$$p(\hat{x}(y) \mid y) \leq p(\arg\max_{x \in X} p(x \mid y) \mid y), \tag{18}$$

the reconstruction that minimizes the error is $\hat{x}(y) = \arg\max_{x \in X} p(x \mid y)$. However, this is theoretical because in most cases $p(x \mid Y)$ is unknown.

## A.2 PROOF 2

We have:

$$\log(1 - \mathcal{E}) = \log\left(\int_{\mathcal{Y}} p(y)(1 - \varepsilon(y)) \, \mathrm{d}y\right) \tag{19}$$

$$= \log\left(\int_{\mathcal{Y}} p(y)p(\hat{x}(y) \mid y)\right) \mathrm{d}y \tag{20}$$

$$\geq \int_{\mathcal{Y}} p(y) \log(p(\hat{x}(y) \mid y)) \, \mathrm{d}y \tag{21}$$

$$= \int_{\mathcal{Y}} p(y) \int_{\mathcal{X}} p(x \mid y) \log(p(\hat{x}(y) \mid y)) \, \mathrm{d}x \, \mathrm{d}y \tag{22}$$

$$\geq \int_{\mathcal{Y}} p(y) \int_{\mathcal{X}} p(x \mid y) \log(p(x \mid y)) \, \mathrm{d}x \, \mathrm{d}y \tag{23}$$

$$= -H(X \mid Y). \tag{24}$$

The Eq. (21) is obtained using Jensen inequality and Eq. (23) is obtained using the result of Proof 1.

Thus,

$$\mathcal{E} \leq 1 - 2^{-H(X|Y)}. \tag{25}$$

## B DEVELOPMENT OF SHADE LOSS

Below is the detail of the development of Equation (8).

$$\mathbb{V}\mathrm{ar}(Y \mid Z) = \int_{\mathcal{Z}} p(z) \mathbb{V}\mathrm{ar}(Y \mid z) \, \mathrm{d}z \tag{26}$$

$$= \int_{\mathcal{Z}} p(z) \int_{\mathcal{Y}} p(y \mid z)(y - \mathbb{E}(Y \mid z))^2 \, \mathrm{d}y \, \mathrm{d}z \tag{27}$$

$$= \int_{\mathcal{Y}} \int_{\mathcal{Z}} p(z)p(y \mid z)(y - \mathbb{E}(Y \mid z))^2 \, \mathrm{d}y \, \mathrm{d}z \tag{28}$$

$$= \int_{\mathcal{Y}} \int_{\mathcal{Z}} p(y)p(z \mid y)(y - \mathbb{E}(Y \mid z))^2 \, \mathrm{d}y \, \mathrm{d}z \tag{29}$$

$$= \int_{\mathcal{Y}} p(y) \int_{\mathcal{Z}} p(z \mid y)(y - \mathbb{E}(Y \mid z))^2 \, \mathrm{d}y \, \mathrm{d}z. \tag{30}$$

## C SHADE GRADIENTS

Here is studied the influence of SHADE on a gradient descent step for a single neuron $Y$ of a single layer and for one training sample $X$. The considered case of a linear layer, we have: $Y = \mathbf{w}^\top X + b$.

The gradient of $\Omega_{\mathrm{SHADE}}$ with respect to $\mathbf{w}$ is:

$$\nabla_{\mathbf{w}} \Omega_{\mathrm{SHADE}} = (\delta_1 + \delta_2)\mathbf{x}$$
$$\text{with } \delta_1 = \sigma(y)(1 - \sigma(y))\big((y - \mu^1)^2 - (y - \mu^0)^2\big)$$
$$\text{and } \delta_2 = 2\sigma(y)(y - \mu^1) + 2(1 - \sigma(y))(y - \mu^0).$$

We can interpret the direction of this gradient by analyzing the two terms $\delta_1$ and $\delta_2$ as follows:

- $\boldsymbol{\delta_1}$: If $(y - \mu^0)^2$ is bigger than $(y - \mu^1)^2$ that means that $y$ is closer to $\mu^1$ than it is to $\mu^0$. Then $\delta_1$ is positive and it contributes to increasing $y$ meaning that it increases the probability of $Z$ being from mode 1. In a way it increases the average margin between positive and negative detections. Note that if there is no ambiguity about the mode of $Z$ meaning that $\sigma(y)$ or $1 - \sigma(y)$ is very small then this term has negligible effect.
- $\boldsymbol{\delta_2}$: This term moves $y$ toward the $\mu^z$ of the mode that presents the bigger probability. This has the effect of concentrating the outputs around their expectancy depending on their mode to get sparser activation.

## D    EXPERIMENTS DETAILS ON MNIST-M

**Dataset splits and creation.**    To create MNIST-M, we kept the provided splits of MNIST, so we have 55,000 training samples, 5,000 validation samples, and 10,000 test samples. Each digit of MNIST is processed to add color and texture by taking a crop in images from BST dataset. This procedure is explained in Ganin & Lempitsky (2015).

**Subsets of limited size.**    To create the training sets of limited size $N$, we keep $N/10$ (since there are 10 classes) randomly picked samples from each class. When increasing $N$ we keep the previously picked samples so the training samples for $N = 100$ are a subset of the ones for $N = 250$. The samples chosen for a given value of $N$ are the same across all models trained using this number of samples.

**Image preprocessing.**    The only preprocessing applied to the input images is that their values are rescaled from $[0, 1]$ to $[-1, 1]$.

**Optimization.**    For the training, we use mini-batch of size 50 and use Adam optimizer with the recommended parameters, *i.e.* $\lambda_r = 0.001, \beta_1 = 0.9, \beta_2 = 0.999, \epsilon = 10^{-8}$.

**Hyperparameter tuning.**    For weight decay and SHADE, the optimal regularization weight of each model (for each value of $N$) has been chosen to maximize the accuracy on the validation sets. We tried the values $\{10^{-i}, i = 1..7\}$.

**Model architecture.**    The model have the following architecture:

- 2D convolution ($64 \times 5 \times 5$ kernel, padding 2, stride 1) + ReLU
- MaxPooling $2 \times 2$
- 2D convolution ($64 \times 3 \times 3$ kernel, padding 1, stride 1) + ReLU
- MaxPooling $2 \times 2$
- 2D convolution ($64 \times 3 \times 3$ kernel, padding 1, stride 1) + ReLU
- MaxPooling $2 \times 2$
- Fully connected (1024 inputs, 10 outputs) + SoftMax

## E    EXPERIMENTS DETAILS ON IMAGENET

The fine tuning in the experiment section 4.2 has been done with momentum-SGD with a learning rate of $10^{-5}$ and a momentum of 0.9 and a batch size of 16 images. It took 8 epochs to converge.

