# OpenReview forum: "SHADE: SHAnnon DEcay Information-Based Regularization for Deep Learning"
_ICLR.cc/2018/Conference — Reject_

### Official Review · AnonReviewer2 · 2017-11-27
**an obvious idea supported by flawed reasoning**

**Rating:** 5
**Confidence:** 4

**Review:**

This paper proposes another entropic regularization term for deep neural nets. The key idea can be stated as follows: Let X denote the observed input, C the hidden class label taking values in a finite set, and Y the representation computed by a neural net. Then C -> X -> Y is a Markov chain. Moreover, assuming that the mapping X -> Y is deterministic (as is the case with neural nets or any other deterministic representations), we can write down the mutual information between X and Y as

I(X;Y) = H(Y) - H(Y|X) = H(Y).

A simple manipulation shows that H(Y) = I(C;Y) + H(Y|C). The authors interpret the first term, I(C;Y), as a data fit term that quantifies the statistical correlations between the class label C and the representation Y, whereas the second term, H(Y|C), is the amount by which the representation Y can be compressed knowing the class label C. The authors then propose to 'explicitly decouple' the data-fit term I(C;Y) from the regularization penalty and focus on minimizing H(Y|C). In fact, they replace this term by the sum of conditional entropies of the form H(Y_{i,k}|C), where Y_{i,k} is the activation of the ith neuron in the kth layer of the neural net. The final step is to recognize that the conditional entropy may not admit a scalable and differentiable estimator, so they use the relation between a quantity called entropy power and second moments to replace the entropic penalty with the conditional variance penalty Var[Y_{i,k}|C]. Since the class-conditional distributions are unknown, a surrogate model Q_{Y|C} is used. The authors present some experimental results as well.

However, this approach has a number of serious flaws. First of all, if the distribution of X is nonatomic and the mapping X -> Y is continuous (in the case of neural nets, it is even Lipschitz), then the mutual information I(X;Y) is infinite. In that case, the representation of I(X;Y) in terms of entropies is not valid -- indeed, one can write the mutual information between two jointly distributed random variables X and Y in terms of differential entropies as I(X;Y) = h(Y) - h(Y|X), but this is possible only if both terms on the right-hand side exist. This is not the case here, so, in particular, one cannot relate I(X;Y) to I(C;Y). Ironically, I(C;Y) is finite, because C takes values in a finite set, so I(C;Y) is at most the log cardinality of the set of labels. One can start, then, simply with I(C;Y) and express it as H(C) - H(C|Y). Both terms are well-defined Shannon entropies, where the first one does not depend on the representation, whereas the second one involves the representation. But then, if the goal is to _minimize_ the mutual information between I(C;Y), it makes sense to _maximize_ the conditional entropy H(C|Y). In short, the line of reasoning that leads to minimizing H(Y|C) is not convincing. Moreover, why is it a good idea to _minimize_ I(C;Y) in the first place? Shouldn't one aim to maximize it subject to structural constraints on the representation, along the lines of InfoMax?

The next issue is the chain of reasoning that leads to replacing H(Y|C) with Var[Y|C]. One could start with that instead without changing the essence of the approach, but then the magic words "Shannon decay" would have to disappear altogether, and the proposed method would lose all of its appeal.

---

> ### Author Response · Authors · 2017-12-21
> **Reply to reviewer 2**
>
> Thanks for your input, however, we believe that your review is hasty since you only base your criticism on wrong assumptions about our paper (among others, you say “the distribution of X is nonatomic” while it is atomic; and “why is it a good idea to _minimize_ I(C;Y) in the first place” while we say the opposite in the introduction) and end with a very aggressive and unfounded comment about our illegitimate use of Shannon entropy. You will find our detailed answers below.
>
> > “If the distribution of X is nonatomic and the mapping X -> Y is continuous [...] then the mutual information I(X;Y) is infinite. In that case, the representation of I(X;Y) in terms of entropies is not valid. [...]”
>
> In fact, for images, X is in a finite input space taking values from {0, 1, 2, …, 255}^(H*W*3). H being the height of the image and W its width. X is therefore atomic and this whole remark does not stand.
>
> > “if the goal is to _minimize_ the mutual information between I(C;Y), it makes sense to _maximize_ the conditional entropy H(C|Y). In short, the line of reasoning that leads to minimizing H(Y|C) is not convincing. Moreover, why is it a good idea to _minimize_ I(C;Y) in the first place? Shouldn't one aim to maximize it subject to structural constraints on the representation, along the lines of InfoMax?”
>
> We never claim that we want to minimize I(C;Y). In fact it is important to get this value as high as possible, because we want to be able to determine C using Y to classify the sample correctly. Following the IB framework the value that we intend to minimize is I(X,Y) (= I(C,Y) + H(Y|C)). However in order not to deteriorate I(C,Y) during the minimization of I(X,Y) we focus on the decoupled term H(Y|C). Moreover we show that the term H(Y|C) is directly related to the invariance of the model and is interesting to be minimized.
>
> > “The next issue is the chain of reasoning that leads to replacing H(Y|C) with Var[Y|C]. One could start with that instead without changing the essence of the approach, but then the magic words "Shannon decay" would have to disappear altogether, and the proposed method would lose all of its appeal.”
>
> Having regularizations that can be theoretically interpreted is important in order to understand its effect and to be able to improve and adapt it. We do not think the appeal of our method comes from the magic word “Shannon” and this unfounded comment along with the title of the review “an obvious idea supported by flawed reasoning” shows a clearly aggressive and biased review.

---

> > ### Comment · AnonReviewer2 · 2017-12-26
> > **insisting on discreteness of the distribution of X completely invalidates the main claim of the paper**
> >
> > > we believe that your review is hasty since you only base your criticism on wrong assumptions about our paper (among others, you say “the distribution of X is nonatomic” while it is atomic ...
> >
> > Actually, my assumption that the distribution of X is nonatomic gave you the maximum benefit of the doubt. If the distribution of X is atomic, then so are the distributions of the activations of all the neurons in the net, for any choice of the weights. But, in that case, the variance bound of Eq. (6) is not valid because it holds only for distributions with well-defined differential entropies, i.e., precisely those distributions that are nonatomic. The variance-based upper bound does not apply to the usual Shannon entropy. So, by insisting that I have misread your paper, you are showing an even bigger lack of understanding of information theory than I had originally imagined.
> >
> > > However in order not to deteriorate I(C,Y) during the minimization of I(X,Y) we focus on the decoupled term H(Y|C).
> >
> > By definition, I(C;Y) = H(Y) - H(Y|C). Hence, I(C;Y) and H(Y|C) are anything but decoupled.
> >
> > I rest my case.

---

> > > ### Author Response · Authors · 2017-12-29
> > > **Reply**
> > >
> > > - Regarding the variance bound:
> > > We do not claim that the inequality holds for any neural network layer output as we point out just below eq 6 "this bound holds for any continuous distributions". However a common assumption in computer vision is that the input space variable is a quantization of an underlying continuous distribution, justifying the use of the variance as an objective function in our optimization problem.
> > >
> > > - Regarding the word "decoupled":
> > > As you surely understood, our use of the word "decoupled" is informal and means that we keep H(Y|C) because when minimized it won't minimize the term I(Y,C), unlike when minimizing I(X,Y). Sure, they are still coupled, but in the opposite direction of the initial one that was problematic. This is really being picky about a word that is here to simply explain the intuition. But sure, we can change the phrasing to be more formally correct.
> > >
> > > In general, we regret the fact that you refuse to discuss the main ideas, originality, development and experiments of our paper; and choose instead to focus on irrelevant and not constructive details, do not admit any misreading on your part (like asking why we want to minimize I(Y,C)) and keep with the aggressive tone ("an even bigger lack of understanding of information theory than I had originally imagined")

---

### Official Review · AnonReviewer3 · 2017-11-28
**nice and intuitive idea**

**Rating:** 7
**Confidence:** 3

**Review:**

the paper adapts the information bottleneck method where a problem has invariance in its structure. specifically, the constraint on the mutual information is changes to one on the conditional  entropy. the paper involves a technical discription how to develop proper estimators for this conditional entropy etc.

this is a nice and intuitive idea. how it interacts with classical regularizers or if it completely dominates classical regularizers would be interesting for the readers.

---

> ### Author Response · Authors · 2017-12-21
> **Reply to Reviewer 3**
>
> Thank you for your review. Regarding the relation with usual regularizers would be interesting. Complementary study in this direction could indeed be the subject of future work.
> As of now, we already have two points in that direction:
> - In Sec. 3 we present a link between a baseline of our regularizer (H(Y) instead of H(Y|C)) and weight decay.
> - In the experiment in Sec 4.1 we empirically study the complementarity of SHADE and Dropout.

---

### Official Review · AnonReviewer1 · 2017-11-29

**Rating:** 4
**Confidence:** 3

**Review:**

Summary:

The paper presents an information theoretic regularizer for deep learning
algorithms. The regularizer aims to enforce compression of the learned
representation while conditioning upon the class label so preventing the
learned code from being constant across classes. The presentation of the Z
latent variable used to simplify the calculation of the entropy H(Y|C) is
confusing and needs revision, but otherwise the paper is interesting.

Major Comments:

- The statement that I(X;Y) = I(C;Y) + H(Y|C) relies upon several properties
  of Y which are not apparent in the text (namely that Y is a function of X,
so I(X;Y) should be maximal, and Y is a smaller code space than X so it should
be H(Y)). If Y is a larger code space than X then it should still be true, but
the logic is more complicated.

- The latent code for Z is unclear. Given the use of ReLUs it seems like Y
  will be O or +ve, and Z will be 0 when Y is 0 and 1 otherwise, so I'm
unclear as to when the value H(Y|Z) will be non-zero. The data is then
partitioned within a batch based on this Z value, and monte carlo sampling is
used to estimate the variance of Y conditioned on Z, but it's really unclear
as to how this behaves as a regularizer, how the z is sampled for each monte
carlo run, and how this influences the gradient. The discussion in Appendix C
doesn't mention how the Z values are generated.

- The discussion on how this method differs from the information bottleneck is
  odd, as the bottleneck is usually minimising the encoding mutual information
I(X;Y) minus the decoding mutual information I(Y;C). So directly minimising
H(Y|C) is similar to the IB, and also minimising H(Y|C) will affect I(C;Y) as
I(C;Y) = H(Y) - H(Y|C).

- The fine tuning experiments (Section 4.2) contain no details on the
  parameters of that tuning (e.g. gradient optimiser, number of epochs,
batch size, learning rates etc).

- Section 4.4 is obvious, and I'd consider it a bug if regularising with label
  information performed worse than regularising without label information.
Essentially it's still adding supervision after you've removed the
classification loss, so it's natural that it would perform better. This
experiment could be moved to the appendix without hurting the paper.

- In appendix A an upper bound is given for the reconstruction error in terms
  of the conditional entropy. This bound should be related to one of the many
upper bounds (e.g. Hellman & Raviv) for the Bayes rate of a predictor, as
there is a fairly wide literature in this area.

Minor Comments:

- The authors do not state what kind of input variations they are trying to
  make the model invariant to, and as it applies to CNNs there are multiple
different kinds, many of which are not amenable to a regularization based
system for inducing invariance.

- The authors should remind the reader once that I(X;Y) = H(Y) - H(Y|X) = H(X) -
  H(X|Y), as this fact is used multiple times throughout the paper, and it may
not necessarily be known by readers in the deep learning community.

- Computing H(Y|C) does not necessarily require computing c separate
  entropies, there are multiple different approaches for computing this
entropy.

- The exposition in section 3 could be improved by saying that H(X|Y) measures
  how much the representation compresses the input, with high values meaning
large amounts of compression, as much of X is thrown away when generating Y.

- The figures are difficult to read when printed in grayscale, the graphs
  should be made more readable when printed this way (e.g. different symbols,
dashed lines etc).

- There are several typos (e.g. pg 5 "staking" -> "stacking").

---

> ### Author Response · Authors · 2017-12-21
> **Reply to Reviewer 1**
>
> Thank you for your feedback. In your review, we noticed one important comment questioning the fundamental difference between our model and IB. However, we noticed a misunderstanding concerning many sections of the development of our SHADE modeling. We try to clarify these points in our answers below. We also made the corresponding changes in the paper to rephrase some paragraphs and improve their clarity (mostly the latent description on page 3 and Sec 4.4).
>
> # Regarding the link with IB
>
> > “[...] how this method differs from the [IB] is odd [...] minimising H(Y|C) is similar to the IB, and minimising H(Y|C) will affect I(C;Y) as I(C;Y) = H(Y) - H(Y|C)”
>
> The IB framework propose to minimize I(X,Y) at constant I(Y,C). Shamir et al. (2010) propose to use I(X;Y) as regularization criterion. However, in the development of I(X,Y) = H(Y) = I(Y,C) + H(Y|C) we identify the term I(Y,C) that we do not want to minimize (but indirectly maximize it since it corresponds to a sort of classification loss), so minimizing I(X,Y) would impact I(Y,C) in an undesired and uncontrolled way. This is why we only minimize the second term H(Y|C) (that will affect I(C,Y) in the desired way by maximizing it). This is mentioned in the introduction and in Sec. 3.
>
> # Other comments about the details of the method
>
> > “I(X;Y) = I(C;Y) + H(Y|C) relies upon several properties of Y which are not apparent [...].
> The only required properties for the development of this equality are described in paragraphs 2 and 4 of the introduction: “Considering an input variable X, label C and its deep representation *Y = h(X)*, IB regularizes the training by minimizing the mutual information I(X, Y) at constant mutual information I(C, Y). [...] For a *deterministic* model, we have I(X, Y) = I(C, Y) + H(Y | C).” The detailed development is presented thoroughly in Sec. 3. The fact that H(Y|X) = 0 in that case has been added in the introduction.
>
> > “The latent code for Z is unclear. Given the use of ReLUs it seems like Y will be O or +ve, and Z will be 0 when Y is 0 and 1 otherwise, so I'm unclear as to when the value H(Y|Z) will be non-zero.
> > “[...] monte carlo sampling is used to estimate the variance of Y conditioned on Z, but it's really unclear as to how this behaves as a regularizer, how the z is sampled [...], how this influences the gradient.”
>
> The intuition behind the variable Z, is that a neuron is responsible for detecting the presence of an attribute on the picture (e.g. the presence of a wheel, Z corresponding to the variable “there is a wheel” which is binomial). The value Y of the neuron *before* the ReLU represents the confidence in the detection. If its activation is high it is very likely that the attribute is present on the picture (Y >> 0 ↔ Z = 1), if it is low it is likely that the attribute is absent (Y << 0 ↔ Z = 0).
>
> Z is a random variable of chosen distribution (below equation 9) that depends on Y but not a deterministic mapping of Y, that is why H(Z|Y) is non-zero. Since we know the distribution p(Z|Y) we can compute everything without sampling on Z, cf Eq. 10 and Algorithm 1. Appendix C describe how the regularization affects the gradients without the need for sampling.
>
> > “Section 4.4 is obvious, and I'd consider it a bug if regularising with label information performed worse than regularising without label information”
>
> We do not use any labels for this as indicated in the paper: “Without the classification loss, the network performance obviously declines as we do not provide information about the labels”. We apply only regularizers, which are Var(Y) and Var(Y|Z). This is here to illustrate the fact that our regularizer is less aggressive toward class information and the importance of conditional entropy described in Sec. 3.
>
> # Comments about minor issues
>
> Regarding the fine tuning experiments, we did not include the details of the training because they are very standard (SGD, lr=1e-5, batches of 16, 8 epochs). They are now in appendix E.
>
> Regarding appendix A and its comparison to the literature, it is simply a theoretical illustration of the discussion in Sec. 3 and not a real contribution, therefore its comparison to the literature is far from the scope of our paper.
>
> Regarding the kind of variations we want to be invariant to, indeed no input variations are specified in this paper as we do not target invariance to particular transformation. In fact the information theoretic framework enables to be agnostic to the type of transformation that is managed by the mapping functions which is a advantage our case. Indeed, modeling rigorously the transformations that the models should be invariant to is very difficult.
>
> Regarding the computation of H(Y|C), there is in fact many ways to estimates a conditional entropy, but few ones fit the gradient descent methodology without making strong assumptions about the distributions.

---

> > ### Comment · AnonReviewer1 · 2017-12-21
> > **Latent coding**
> >
> > So the latent coding Z relies upon the network using ReLUs? As if you model Z as sigmoid(Y) then using a sigmoid activation function will reduce that operation to the identity. This leads to a further thought, which is that the SHADE regulariser is essentially incorporating information about how negative the value was before the ReLU, and so isn't it a form of leaky ReLU? Given how closely this regulariser appears tied to the activation function, it would benefit the paper to compare against other approaches which try to improve upon the ReLU.
> >
> > The text in section 2.3 does not make explicit the switch from a variance estimated using monte carlo, to a deterministic estimator, so could do with a little revision to make this clear.
> >
> > Also in the comment about Section 4.4 the paper states it did not use any labels, but the text in section 4.4 talks about how this regulariser incorporates label information by modelling it with Z. Hence my comment in the original review about a regulariser that knows of the existence of labels performing better than one without.

---

> > > ### Author Response · Authors · 2017-12-27
> > > **Reply**
> > >
> > > The latent coding doesn't require the use of ReLU despite there is a clear link  between SHADE intuition and the popular activation functions. Precisely, the fact that a neuron (pre-activation) encodes a binomial information (active or not), with a (soft) threshold at 0, which is coherent with the idea behind all usual activation functions (ReLU, sigmoid, ...) used with neural networks. Advanced activation functions like LeakyReLU try to adjust how gradient is backpropagated while SHADE only applies a layerwise regularization on the weights according to the latent variable. Both have different goals but it would be interesting, indeed, to study the impact of different activation functions, such as LeakyReLU, on the latent variable distribution.
> > >
> > > In section 2.3, the monte carlo sampling approximating (8) with (9) is done on the variable Y over \mathcal{Y} only. It simulates the distribution of the activations Y for the input data distribution on \mathcal{X}. We therefore apply monte-carlo with mini-batches sampling, i.e. the same thing that is done on any loss to optimize a neural net. Remains an integral over \mathcal{Z} which becomes a sum over the 2 modes of Z (Z=0, Z=1) and the expectancy is estimated with a moving average. We are going to make this fact clearer in the paper.
> > >
> > > We argue that the class information is implicitely contained in the latent variable Z. This point is not obvious. This is why we are trying to demonstrate it in experiment 4.4, using the fact that after training, each layer contains enough class information (with a forward pass) to predict the labels with a good accuracy. However it is possible that this information is not accurate enough to predict the correct class (the label in the case of training). In fact the training brings the latent variable closer to a sufficient statistic of X for C, making it more powerful to predict the class.

---

### Official Review · AnonReviewer4 · 2018-01-13
**Some nice ideas but also some tenuous connections**

**Rating:** 4
**Confidence:** 3

**Review:**

The authors propose a particular variance regularizer on activations and connect it to the conditional entropy of the activation given the class label. They also present some competitive results on CIFAR-10 and ImageNet.

Despite some promising results, I found some issues with the paper. The main one is that the connection between conditional entropy and the proposed variance regularizer seems tenuous. The chain of reasoning is as follows:

- Estimation of H(Y|C) is difficult for two reasons: 1) when the number of classes is large, the number of samples needed calculate the entropy are high, and 2) naive estimators -- even when the number of classes are small -- have high variance. To solve these issues, the authors propose:

a) Introduce a latent code Z such that H(Y|C) = H(Y|Z). This solves problem 1).

b) Use a variance upper bound on H(Y|Z). This solves problem 2).

My issue is with the reasoning behind a). H(Y|C) = H(Y|Z) relies on the assumption that I(Y;C) = I(Y;Z). The authors present a plausibility argument, but the argument was not sufficiently convincing to me to overcome my prior that I(Y;C) =/= I(Y;Z).

Apart from this, I found some other issues.

* In the second paragraph of 2.2, the acronym LME in "LME estimator" was not defined, so I checked the reference provided. That paper did not mention a LME estimator, but did present a "maximum likelihood estimator" with the same convergence properties as those mentioned in the SHADE paper. Since the acronym LME was used twice, I'm assuming this was not a typo. Perhaps this is a bug in the reference?

* In section 4.4, it's hard to know if the curves actually show that "SHADE produces less class-information filtering". The curves are close throughout and are nearly identical at epoch 70. It is entirely possible that the difference in curves is due to optimization or some other lurking factor.

* The final form of the regularization makes it look like a more principled alternative to batchnorm. It would have been nice if the authors more directly compared SHADE to BN.

* There are two upper bounds here: that H(Y_l | C) <= \sum_i H(Y_{l, i} | C), and the variance upper bound. The first one does not seem particularly tight, especially at the early layers where the representation is overcomplete. I understand that the authors argue that the upper bound is tight in footnote 3, but it is only plausible for later laters.

My Occam's razor explanation of this paper is that it forces pre-nonlinearity activations (and hence post-nonlinearity activations) to be binary, without having to resort to sigmoid or tanh nonlinearities. This is a nice property, but whether the regularizer connects to H(Y|C) still remains unsolved.

---

### Decision · Program_Chairs · 2018-01-29
**ICLR 2018 Conference Acceptance Decision**

**Decision:**

Reject

**Comment:**

The proposed conditional variance regularizer looks interesting and the results show some promise. However, as the reviewers pointed out, the connection between the information-theoretic argument provided and the final form of the regularizer is too tenuous in its current form. Since this argument is central to the paper, the authors are urged to either provide a more rigorous derivation or motivate the regularizer more directly and place more emphasis on its empirical evaluation.